# Neuroendocrine Neoplasms of the Gastrointestinal Tract versus Neuroendocrine Neoplasms of the Gynaecological Tract—Comparison of the Risk Factors and Non-Surgical Treatment Efficacy

**DOI:** 10.3390/ijms24076853

**Published:** 2023-04-06

**Authors:** Anna Lorenz, Sebastian Lenkiewicz, Mateusz Kozłowski, Sebastian Kwiatkowski, Aneta Cymbaluk-Płoska

**Affiliations:** 1Department of Reconstructive Surgery and Gynecological Oncology, Pomeranian Medical University in Szczecin, Al. Powstańców Wielkopolskich 72, 70-111 Szczecin, Poland; 2Department of Obstetrics and Gynecology, Pomeranian Medical University in Szczecin, Al. Powstańców Wielkopolskich 72, 70-111 Szczecin, Poland

**Keywords:** neuroendocrine tumours, gastrointestinal tract, reproductive tract

## Abstract

Neuroendocrine tumours of the gastrointestinal tract are rare. The incidence has increased in recent years due to improvements in diagnostic methods for detecting these lesions. These tumours have a poor prognosis, especially when detected at an advanced stage. The basis of the treatment is resection, and non-surgical treatments are also standard in the treatment process. The situation is similar in even rarer neuroendocrine tumours of the reproductive tract, which are associated with an equally poor prognosis. In this article, we focus on learning about the risk factors (including genetic mutations) that increase the risk of the disease and comparing the effectiveness of non-surgical treatments—chemotherapy, radiotherapy, peptide receptor radionuclide therapy, somatostatin analogues, and immunotherapy. The efficacy of these treatments varies, and immunotherapy appears to be a promising form of treatment; however, this requires further research.

## 1. Introduction

Neuroendocrine neoplasms are a group of heterogeneous tumours. They arise from mutant neuroendocrine cells that are mainly located in the mucosa and the submucosa. Histologically, they originate in the neuroectoderm, endoderm, and neural crest [1]. They synthesise and secrete (or store) chemical messengers such as neuropeptides and amines. It is from these cells that substances, such as gastrin, cholecystokinin, insulin, ghrelin, vasoactive intestinal factor, motilin, secretin, and others, originate. These cells also express markers, including chromogranin A and synaptophysin [2]. They are relatively rare. In the United States, the incidence rate has changed over the years. In 1973, it was 1.09/100,000 to 6.89/100,000 in 2012. Incidence rates in organs were as follows: lung 1.49/100000, gastrointestinal tract with pancreas 3.56/100,000, and neuroendocrine tumours of unknown initial location 0.84/100,000 [3]. In England, the incidence rate for the gastrointestinal tract increased from 0.27/100,000 per year for men in 1971 to 1.32/100,000 in 2006. For women in the same years, the situation is similar—from 0.35 to 1.33/100,000 per year. The appendix appeared to be the most commonly affected organ, with 38% of cases [4]. In Germany, Canada, and Japan, a similar increase in incidence has been noted [5,6,7]. The reason is attributed to better diagnostics, including imaging. The 2017 WHO classification [8] divides NENs into neuroendocrine tumours (NETs) and neuroendocrine carcinomas (NECs). NETs are graded into three types: G1, G2, and G3. Grading depends on proliferation activity, as measured by the mitotic index and/or Ki-67 index. NEC are, by definition, poorly differentiated lesions of high grade. A distinction is made between small- and large-cell NEC. Various methods are used to diagnose GEP NENs. One is the assessment of serum chromogranin A and B levels; however, this method is not specific. Its levels can be elevated in a variety of other conditions, not only in GEP NENs [2,9]. Various techniques are used in diagnostic imaging—ultrasound, endoscopy, and computed tomography. One of the more sensitive imaging methods is 68Ga-labeled somatostatin analogues CT/PET. As is well known, diagnostic imaging contributes not only to the detection of NETs but also to the staging of the disease [10]. According to ENETS and WHO guidelines [11], histopathological examination of the tumour should be complemented by immunohistochemical examination (evaluation of the Ki-67 index), which allows the tumour to be classified accordingly. The immunohistochemical examination also determines the expression of the granins, chromogranin A, B, and secretogranin II; however, depending on the location of the tumour, their expression may vary significantly [12].

The situation is different for neuroendocrine tumours of the reproductive tract. These are less common than gastrointestinal or respiratory tract tumours, and consequently, there are few guidelines for their treatment. They account for approximately 2% of all malignant neoplasms of the reproductive tract [13,14]. They are most commonly localised in the cervix, followed by the ovary or the endometrium. Less common locations are the vulva and the vagina [13,15]. One must be especially cautious when treating female reproductive tract tumours for various reasons. Not only do these NEN vary from gastrointestinal tract NEN, but what may be equally important is that many of the primary reproductive tract neoplasms tend to be endocrinally active. It is important not to misdiagnose a primary tumour as a NEN since not only are they different regarding biomarkers and its origin, but they might respond to a different second-line treatment such as chemo- or immunotherapy. The WHO classification has changed over the years [16]. In 2014, *The Classification of Neuroendocrine Tumours of the Reproductive Tract* (4th edition) divided tumours into two main groups: low and high malignancy. It was then established that the concept of carcinoma and atypical carcinoma would be assigned as counterparts to neuroendocrine tumours (NETs)—low-grade malignancies. Small- and giant-cell tumours are classified as neuroendocrine carcinomas (NEC). It is important to note that this classification applied to cervical, endometrial, vaginal, and uterine tumours and did not include ovarian tumours. In 2017–2018, a new classification was introduced, which is still in force today. *The WHO Tumour Classification* (5th edition) [17] divided NENs into well-differentiated neuroendocrine tumours (NET, grades 1 and 2) and poorly differentiated neuroendocrine carcinomas (NEC, grade 3). The concept of typical and atypical carcinomas has been withdrawn, with the exception of ovarian tumours. In ovarian tumours, there is also no differentiation into grade 1 and 2 NETs. This can be summarised as follows: ovarian carcinomas are stage 1 tumours, and neuroendocrine ovarian carcinomas are stage 3. Neuroendocrine carcinomas (not only of the ovary) can then be differentiated into small, giant, and mixed (also with other types of malignancy) cells. Staging is determined using the FIGO classification, which from 2018 also includes lymph nodes [18]. Primary ovarian carcinomas are uncommon, accounting for approximately 1% of all carcinomas and 0.1% of all malignant ovarian tumours [19,20]. They can be divided into four types: insular, trabecular, strumal, and mucinous. The most common type is insular, accounting for about 50% of cases [16]. The strumal type usually occurs bilaterally at around 49 years [18]. Interestingly, the cases of strumal tumours producing peptide YY have been described as acts to inhibit gastrointestinal motility and result in constipation [21]. NENs of the reproductive tract, such as GEP NENs, also show the expression of synaptophysin and chromogranin A, which is used in immunohistochemical studies [19]. It is worth noting that these tumours additionally show CD56 expression, but this is not specific to this tumour—it is also found in squamous cell carcinomas and adenocarcinomas [22]. CD56 expression has also been detected in GEP NENs [23]. One diagnostic method is to measure serum chromogranin A levels, but as mentioned earlier, this is not a highly specific method for detecting this type of tumour. The same is true for neurospecific enolase [14]. Diagnostic imaging uses ultrasound, somatostatin receptor imaging (SRI) with various radionuclides and PET and CT scans with radioisotopes, such as 68Gallium. As it turns out, this diagnostic method shows high sensitivity (96%) and specificity (100%) for these tumours [24].

## 2. Risk Factors

Neuroendocrine tumours are rare, so their pathogenesis and risk factors are not fully understood. In 2008, Hassan et al. [25] conducted an analysis of risk factors. They showed that, depending on gender, there is a higher risk of developing particular NETs. For example, for women, there is a higher likelihood of NET tumours of the small intestine (1.1 to 1.0), the stomach (2.4 to 1.0), the lung (1.7 to 1.0), and the rectum (1.3 to 1.0). Race was also included as a risk factor—it is higher for African Americans than Whites. Statistical analysis showed that an age below 60 increases the risk of the disease regardless of gender. Smoking and alcohol consumption do not statistically affect the incidence of this type of cancer, while a positive family history of cancer and diabetes (especially treated with insulin) significantly increases the risk of the disease. Rinzivillo et al. [26] examined risk and protective factors for the NET of the small bowel excluding hereditary family syndromes. They showed that a positive family history of colorectal cancer and breast cancer increased the risk for NET of the small bowel. Alcohol consumption of more than 21 units per week and smoking a significant amount of cigarettes (20 packs of cigarettes or more per year) also increase the risk for NET of the small bowel. Interestingly, taking aspirin at least twice a week for at least a year can be considered a protective factor. Similar conclusions were made by Haugvik et al. [27] for NET of the small bowel. Positive family history of any cancer or colorectal cancer and smoking are risk factors for NET of the small bowel. A risk factor analysis for NET of the pancreas was performed [28]. The diagnosis of diabetes and a family history of certain cancers (e.g., sarcoma or bladder cancer) were shown to be significant risk factors. Smoking did not significantly influence the increased risk of the disease. Alcohol consumption was also ruled out as a risk factor, although the study authors indicated that alcohol consumption requires further research as well as validation of the data. For sporadic pancreatic NETs [29], statistical analysis showed that cigarette smoking, diagnosed diabetes mellitus, and a positive history of grade I cancer were independent risk factors for non-functional tumours, while alcohol abuse contributed to the development of active NET tumours. Rectal NETs [30] have been shown to be most likely to affect patients aged <50 years, which is a much younger age range compared to other studies [25,31]. It appears that a study by Jung et al. [30] found that men are more likely to develop a rectal NET. It has been suggested that in Asian countries, the male gender has a higher risk of developing rectal NET, although the authors indicated that this requires further research as, for example, in the USA [25], women are more likely to suffer from this disease. Drinking alcohol in a significant amount and, interestingly, low HDL-C levels were associated with higher risk. Smoking was excluded as a risk factor. Pyo et al. [32] published their findings showing the following risk factors for a rectal NET—a positive family history of grade I cancer, the presence of metabolic syndrome and high serum cholesterol and ferritin levels. As it turned out, the consumption of red meat and the consequent elevation of ferritin and cholesterol levels increase the risk of metabolic disorders and diabetes [33,34], which raises the risk of rectal NET.

Mutations leading to the development of neuroendocrine tumours are still a focus of research. Puccini et al. [35] performed a molecular analysis of NETs of the gastrointestinal tract and the pancreas. They showed that the frequency of different mutations varied in the gastrointestinal tract and the pancreas. They found that mutations, such as *FOXO3* (8.6%), *MEN1* (25.9%), *ATRX* (20.6%), and *TSC2* (6.3%), were more common in the pancreas, while *APC* (13.8%) was more common in the gastrointestinal tract. When dividing the gastrointestinal tract into upper and lower sections, the frequency of mutations also differed. In the upper segment, *BRCA2* (7.5% vs. 0%), *TP53* (33% vs. 16%), and *CTNNB1* (4.7% vs. 0%) mutations were more frequent, while *APC* mutations were less frequent (4.7% vs. 16%). In addition, mutations in the *KRAS* (29.4%), *TP53* (51%), *PI3KCA* (7%), and *BRAF* (5.4%) genes were also detected in high-grade GEP NENs.

Another study [36] showed that the most common mutations associated with GEP-NEC tumours were *TP53* (57% of cases, poorly differentiated tumours) and *KRAS* (30%). Additionally, *PIK3CA/PTEN* (22%) and *BRAF* (13%) were detected. *TP53* mutation was associated with a worse prognosis, while no genetic alteration was found in the small intestine tumours (Ki67 < 2%). In addition to the aforementioned mutations, *APC* (large intestine), *RB1* (pancreas, large intestine), and *CTNNB1* (large intestine) mutations were also found in poorly differentiated NEC at different locations. As it turned out, the only mutation for poorly differentiated NEC of the small intestine was *TP53*.

Another molecular study [37] divided neuroendocrine tumours into three categories: NET G3, NEC with Ki67 <55%, and NEC with Ki67 ≥55%. *TP53* (32.9%), *KRAS* (5.5%), and *BRAF* (4.1%) mutations were detected in all tumour groups, with NEC ≥55% being more common (76.7%) than in the other two groups (55.6% for NEC <55% and 20% for NET). Mean overall survival was highest for NET (4.3 years), followed by 1.8 years for NEC <55% and only 0.7 years for NEC ≥55%.

In another comparative analysis [38], mutations for pancreatic NETs were extracted, excluding familial syndromes and small- and giant-cell carcinomas. The percentage distribution was as follows: *MEN1* (44.1%), *DAXX* (25%), *ATRX* (17.6%), *PTEN* (7.3%), *TSC2* (8.8%), and *PIK3CA* (1.4%). NET pancreatic mutations and pancreatic ductal adenocarcinoma (PDAC) were also compared. It turned out that PNETs contained approximately 60% fewer mutations per tumour compared to PDAC. The two tumour types also differ in their mutated genes. A C-to-T transversion was more common in PDAC, and a C-to-G transversion was more common in PNET tumours.

Another study of pancreatic NENs [39] analysed 12 cases of small- and giant-cell carcinomas and 11 cases of G3 PNETs. The results showed the presence of *TP53* (67%) and *KRAS* (42%) mutations. *RB1* mutation was detected in only one tumour. In PNET G3, the mutated MEN1 gene (45%) and *TCS2* (27%) were detected. In the remaining cancers and PNETs, multiple mutations were detected in various combinations but in much lower numbers.

In the case of small bowel NET, one of the known mutations is *CDKN1B* and *P27* (KIP1 variant) [40].

Loss of P27 function resulted in a poorer prognosis and higher stage at tumour detection. As we already know [41], MEN4 is caused by a mutation in *CDKN1B*. The *CDKN1B* gene is responsible for the normal production of P27, a tumour suppressor protein that regulates the cell cycle. In addition, the MEN1 gene product (menin) affects the expression of the *CDKN1B* and *CDKN2C* genes by binding to the promoters of these genes.

In conclusion, not all mutations causing NEN of the gastrointestinal tract are known, and this requires further research. The molecular profiles of pancreatic and gastrointestinal NENs differ. Figure 1 summarises the most common pancreatic and gastrointestinal NEN mutations.

For NEN of the reproductive tract, the risk factors are even less well understood. One pathogen that increases the risk of disease is HPV. An analysis [42] of more than 10,000 cases of invasive cervical cancer was performed. NET accounted for 49 cases, representing approximately 0.5% of all cervical cancers. HPV DNA was detected in 85.7% of cases, of which 55% were HPV16 positive, and 41% were HPV18 positive. The remaining 4% were other HPV types. Translating this to all cases of invasive cervical cancer (84.2% HPV positive), the presence of HPV DNA was statistically and significantly higher in NETs. Several other published papers [43,44,45,46] have made similar conclusions about the importance of HPV in the pathogenesis of cervical NENs.

In the NEN of the endometrium, the CA-125 marker may be elevated [16], although this is not a consistently associated phenomenon.

An analysis of high-grade cervical NECs was performed [47]. It revealed that the most frequent mutations were *PIK3CA* (19.6%), *TP53* (15.5%), *MYC* (1.5%), and *PTEN* (14.4%). The patient group was also divided according to HPV infection with 85.6% of patients not being positive and 14.4% negative. In addition to mutations such as *PIK3CA, TP53*, and *PTEN, RB1* and *ARID1A* also appeared. Interestingly, abnormalities within the RAS pathway—mutations in the *KRAS* and *BRAF* genes—were detected in several cases (11 out of 97). In 9 cases out of 97, alterations within the homozygous recombination deficiency (HRD) pathway were detected, the most common alteration being a mutation in the *BRCA2* gene.

In another study [48], the profile of genetic alterations was similar with the most common mutations being *PIK3CA* (17%) and *PTEN* (10%), which appeared with DDL3 expression. Other mutations such as *TP53* (17% of cases), *KRAS* (11%), and *CTNNB1*(5%) lacked association with DDL3 expression. In addition, PD-L1 expression was detected in 10% of cases creating a potent possibility for treatment with immunotherapy. When mutations associated with small-cell cervical NEC were examined [49], similar results were obtained with the presence of *PIK3CA* mutations reported at 18%, *KRAS* at 14%, and *TP53* at 11%.

Another analysis of cervical NEC [50] examined the presence of the HPV infection and the concurrent mutations. As it turned out, p53 (16.7%) and *NRAS* (3.3%) mutations were detected, with *KRAS*, *BRAF*, and *EGFR* mutations excluded. Positive results for HPV were 86.7%. Additionally, overexpression of p16INK4a was detected, which most likely correlates with the HPV infection.

In small-cell ovarian carcinomas [43,51,52] of the hypercalcaemic type, a mutation in the *SMARCA4* gene, resulting in the loss of function of this protein and loss of the expression, increases the risk of ovarian NEC. In addition to mutations in the *SMARCA4* gene, one risk factor is a mutation in the *SMARCB1* gene. The products of these genes affect the SWI/SNF complex, which promotes oncogenesis [53]. However, due to the significant molecular differences between this tumour and NENs and the greater similarity to rhabdoid-like tumours, it appears that this neoplasm probably does not belong to NENs at all [16].

Figure 2 summarises the most common NEN mutations of the reproductive tract.

Briefly summarised, the risk factors for both cancer types are not fully understood. Both GEP NENs and GT NENs often show mutations in genes such as *TP53*, *KRAS*, *PTEN*, and *PIK3CA*. However, GEP NENs are more common than GT NENs, and consequently, more risk factors (smoking and alcohol abuse) and associated mutations are known. Additionally, interestingly, one of the important risk factors for GT NENs is HPV infection.

### Treatment

According to the ESMO guidelines [24] for GEP NENs, surgery is the treatment of choice in NETs G1 and G2 in local or locoregional diseases. For Pan-NETs >2cm, pancreatectomy with regional lymphadenectomy is recommended because of the increased risk of distant metastasis. Caution should be exercised in high-risk tumours (for example, Pan-NETs G3). For localised NETs of the small bowel, macroscopic radical resection with removal of mesenteric lymph nodes is recommended. 

In cases of advanced disease, surgery should be performed with caution. It is not indicated in G3-advanced NEC or the stage of extra-abdominal metastasis. One of the aims of this article was to evaluate the efficacy of non-surgical methods of treating NENs, such as PRRT, SSA, and chemotherapy. One of the aims of this article was to focus on a literature review and evaluate the efficacy of other forms of the treatment than surgical resection.

## 3. Peptide Receptor Radionuclide Therapy

As some studies [54,55] have shown, NEN tumours express somatostatin receptors on their cell surfaces, which offers the possibility of implementing a therapeutic method of PRRT (peptide receptor radionuclide therapy). The radionuclide-labelled drug is then administered to patients. According to ESMO guidelines [24], PRRT can be considered second-line therapy in case of failure of somatostatin analogues, possibly in Pan-NETs after other therapeutic approaches have been used. 

For GEP-NENs, several studies have been published on this topic. In Nicolini et al. [56], PRRT was administered to 33 patients with GEP-NENs with somatostatin receptors present (SRI+) at the metastatic stage of the tumour. The Ki-67 index ranged from 15–70%. All patients received 4 or 5 cycles of ^177^Lu-DOTATATE drug administration with activity per cycle in the range of 3.7 to 5.5 Bgq. The response to the treatment was presented as follows: 6% of patients showed a partial response, and 64% of the patients had stabilised the disease, giving a complete response rate of 70%. The mean PFS (progression-free survival) varied according to the Ki-67 index, with patients with a Ki-67 ≤35% index averaging 26.3 months and patients with a Ki-67 >35% index averaging 6.8 months, while overall survival (OS) was 52.9 months and 12.6 months, respectively. The authors suggested that a Ki-67% index of 35% was a better indicator than the standard 20%. 

Another study [57] divided patients with well-differentiated NETs and metastases into two groups. The first group received ^177^Lu-Dotatate at a dose of 7.4 Gbq every 8 weeks in combination with octreotide LAR (long-acting repeatable) at a dose of 30 mg intramuscularly, while the second group received octreotide LAR alone. After 20 months, progression-free survival in the 177LuDotatate group was 65.2%, while in the second group, it was only 10.8%. The response rate in the first group was 18%, while in the second group, it was only 3%.

In another study [58], radionuclides were used in GEP NENs G3. Three radionuclides, -^177^Lutetium, ^90^Yttrium, or ^111^Indium, and their combinations were used. Only 30 patients out of 149 had first-line treatment; the rest had second or further treatment. A complete response to the treatment was reported in 1% of patients, 41% had a partial response, 38% achieved disease stabilisation, and 20% experienced disease progression. PFS was 14 months, and OS was 29 months. The Ki-67 index value also influenced prognosis, with a PFS of 55% or greater at 6 months and OS at 9 months. With a value below 55%, PFS was 16 months, and OS was 31 months. 

Another research study also investigated the efficacy of 7.9 GBq ^177^Lu-octreotate radiopeptides with combined chemotherapy in stage 1 or 2 pancreatic NETs [59]. Radiopeptides were administered with capecitabine 1500 mg/m^2^ and temozolomide 200 mg/m2. Complete remission (CR) was demonstrated in 13% of patients and partial remission in 70%. Disease progression was not observed in any patient. The median progression-free survival (PFS) was 48 months. In a previous study [60], similar results were obtained. For NETs of the pancreas without chemotherapeutic agents [61], when radiopeptides were administered, the results presented a slightly different picture. A partial response to the treatment was noted in 60.3% of patients, a lesser response in 11.8%, a disease stabilisation in 13.2%, and a progression in 14.7%. The mean disease-progression-free period was 34 months, and overall survival (OS) was 53 months.

It has been shown in some studies [62] that cervical NENs express somatostatin receptors, potentially providing further therapeutic options, including peptide receptor radionuclide therapy.

## 4. Radiotherapy

Classical radiotherapy is also used as a treatment modality. For primary NETs of the pancreas, which have been shown to be unresectable [63], the results of treatment have been as follows: among NET-G2, the response to treatment was 28.5%, with symptom relief in 33.3% of patients. Among NEC patients, the response to treatment was 25%. The mean PFS for all patients was 5.5 months and OS was 35.9 months. In another study [64] of PNET radiotherapy, similar results were obtained (39% response rate, 26% partial response, 13% complete response, 56% disease stabilisation, and 4% disease progression). Comparing these results to an older work [65], the results appear to be similar. Another study [66] tested the efficacy of adjuvant radiotherapy with the exclusion of adjuvant chemotherapy. The overall 5-year survival was 62%. As it turned out, patients after radiotherapy were less likely to have a local recurrence (6% vs. 10%) compared to the patients not treated with this method, although they were more likely to have distant metastases (38% vs. 18%). The Society of Gynaecologic Oncology [16] suggests in its recommendations that the use of radiotherapy in early-stage cervical NEN is controversial, and the results of studies are contradictory. Many recommendations have been published recommending treatment with radiotherapy in combination with chemotherapy in advanced stages. Dong et al. [67] showed that in a group of patients without distant metastases (M0), there was no significant difference between using surgical treatment alone and combining with radiotherapy, while for distant metastases, the difference was noticeable (44.6 months vs. 80.9 months). Similar results were noted for adjuvant radiotherapy, regardless of the presence of metastases. The efficacy of surgical treatment was compared with EBRT (external beam radiation therapy) [68] alone or in combination with brachytherapy. As it turned out, brachytherapy had no significant effect on survival. In another study [69], the authors suggested no need for radiotherapy even in high-risk patients. In a slightly older study [70], the role of radiotherapy (before and after surgery) seemed to be unclear.

## 5. Somatostatin Analogues

One form of treatment is somatostatin analogues: octreotide and lanreotide. These analogues inhibit tumour growth (they have an antiproliferative effect) and help to reduce hormone secretion [14]. Somatostatin analogues are recommended for the treatment of functional NETs in combination with local therapy. They are also effective in carcinoma syndromes [24].

In a randomised clinical trial [71], patients with well or moderately differentiated GEP NEN were administered lanreotide or placebo. The aim of the study was to test the tumour-suppressive properties of lanreotide. After 24 months, PFS was 65.1% in the lanreotide group and 33% in the placebo group. The researchers noted no significant difference in either group in the quality of life or overall survival. Another study [72] administered long-acting somatostatin analogues, lanreotide autogel or octreotide LAR, to patients diagnosed with NENs (gastrointestinal, thoracic, or other sites). The aim of the study was to evaluate the efficacy according to Ki-67 index values. The response rate for the whole group was 11%, with disease stabilisation in 58% of patients and progression in 31% of patients. PFS was longer in G1 tumours (89 months) than in G2 tumours (43 months) and was 89% in tumours with a Ki-67 index <5%, which is higher than PFS in tumours with a Ki-67 index ≥5% (35 months).

For well-differentiated pancreatic NETs [73], the efficacy of long-acting somatostatin analogues was tested at Ki-67 index values ≥10%. No statistical difference was noted between lanreotide autogel and octreotide LAR. PFS was 11.9 months, and overall survival was approximately 86 months in patients with G2 tumours and 56.8 months in G3 patients. The authors concluded that with Ki-67 index values ≥10%, somatostatin analogues are useful for the treatment of pancreatic NETs. Another study [74] investigated the efficacy of lanreotide autogel and its antiproliferative properties in NENs, most of which were GEP NENs. The mean PFS was 12.9 months. A stabilisation of the disease was seen in 89% of patients and a partial response to the treatment in 4%.

The CLARINET FORTE trial [75] had slightly different results, with NENs divided into two groups—midgut and pancreatic. PFS in the midgut group averaged 8.3 months and 5.6 months in the pancreas group, while with a Ki-67 index value ≤10%, PFS was 8.3 months and 8.0 months, respectively.

The efficacy of lanreotide in well-differentiated GEP NET was also investigated [76]. Patients with a Ki-67 index <10% were included in the study. A partial response to the treatment was reported in 2.2% of patients, while stabilisation of the disease was reported in 88.9% of patients. The mean PFS was 16.4 months.

Based on the above data, it can be concluded that somatostatin analogues have an impact on progression-free survival by stabilising the disease.

In the case of cervical NEN, as previously mentioned, somatostatin receptor expression has been demonstrated [62,77], which offers a potential treatment option. In the case of ovarian-derived carcinoma syndrome, several case reports have been published [78,79,80,81] in which somatostatin analogues have been used, resulting in symptom relief in patients with heart failure.

## 6. Immunotherapy

In the case of GEP NENs, several studies have been published. The most commonly used antibodies are anti-PD-L1 antibodies such as pembrolizumab, nivolumab, and, more recently, spartalizumab. According to some studies, a subset of GEP NENs expresses PD-L1 [82,83,84], with results varying widely (from 10% to up to 70%).

The KEYNOTE-028 clinical trial [85] analysed PD-L1 expression in the metastatic carcinoids and the well and moderately differentiated NETs of the pancreas. In carcinoids, 21% of patients showed PD-L1 expression, whereas, in the pancreatic NET group, 25% did. The patients were given pembrolizumab at a dose of 10 mg/kg every fortnight for 2 years. The response rate was 12% and 6.3%. Some patients showed a partial response to the treatment, but none achieved a complete response.

The KEYNOTE-158 study [86] investigated the efficacy of pembrolizumab in NETs of the lung, the pancreas, the small and large intestines, and the appendix. PD-L1 expression was detected in 15.9% of the tumours. The overall response rate was 3.7% in patients with PD-L1 negative tumours (no complete response and 4 partial responses). The mean OS was 24.4 months, and the PFS was 4.1 months.

MacFarlane et al. [87] investigated the efficacy of pembrolizumab in extrapancreatic G3 NENs. Pancreatic NENs accounted for 29% of cases, while those from the gastrointestinal tract accounted for 64% and 6% with an unknown primary location. Only 4.8% of patients showed a partial response to the treatment, and 14.3% showed a stabilisation of the disease. The mean PFS for this study was approximately 2 months (60 days to be precise).

In another study investigating the same tumour types (NEN G3 of the gastrointestinal tract and the pancreas) [88], the results were similar, with no significant difference between the overall survival, the progression-free period, and the disease control rate in tumours with and without PD-L1 expression.

In a recently published study [89], 24 patients were administered sintilimab at different doses and regimens, of whom 7 were diagnosed with pancreatic NET and 10 with gastrointestinal (GI) NET. The response rate for the whole group was 20.8% and 27.8% for the NEC group. The mean progression-free survival was 2.2 and 2.1 in NET and NEC, respectively. Interestingly, PD-L1 expression was 18.8% and did not correlate with the treatment response. Another study investigated the efficacy of sintilimab in well-differentiated NET (including the pancreas and gastrointestinal tract) and poorly differentiated GEP NEC. The results were as follows (for NET and NEC, respectively): a partial response rate of 7.4 vs. 4.8%, a disease stabilisation rate of 55.8% vs. 14.3%, an unknown response rate of 6.3 vs. 14.3%, and a disease control rate of 63.2% vs. 19.%.

The treatment of extrapancreatic NETs with the combination of anti-CTLA4 (ipilimumab) and anti-PD-L1 (nivolumab) antibodies [90] resulted in an overall response rate of 25%. When tumours were subdivided into high-grade NEC and low- and intermediate-grade NET, the overall response rates reached 44% and 0%, respectively. PFS at 6 months for the entire group was 31%, and the overall survival was 11 months. Using the same treatment in another study [91], similar results were obtained.

Toripalimab, a humanised IgG4 class antibody against the PD-1 receptor, was also used in the treatment of GEP NENs [92]. The response to the treatment reached 20%, and the disease response rate reached 35%. Patients with positive PD-L1 expression had a better response to the treatment compared to negative patients (50% vs. 10.7%). Interestingly, the *ARID1A* mutation was detected in 37.5% of patients.

The above data are briefly summarised in a table (Table 1). Immunotherapy appears to be a promising form of treatment; however, this requires further investigation. Perhaps in the future, it will become a highly effective treatment method.

In the case of NENs of the reproductive tract, several studies have been published.

Frumovitz et al. [93] conducted an analysis of the efficacy of the treatment of NENs of the lower genital tract in seven patients. They were administered intravenous pembrolizumab every three weeks. A patient had a stable disease that started to deteriorate after 14 weeks, and the others had a progression during the entire administration of the drug. The mean time without progression was 2.1 months. One can conclude from this study that pembrolizumab is not very effective, although the small study group must be taken into consideration. In a case report [94] of small cell carcinoma of the cervix, pembrolizumab was used in the treatment. During the treatment, there was a rapid progression of the disease confirmed radiologically. The authors detected an AKT1 E17K mutation and suggested that this may be responsible for the failure of the treatment; however, this requires further investigation. In another single case [95] of small cell carcinoma of the cervix, the patient was given nivolumbium, which proved to be an effective treatment; the disease did not recur. Interestingly, the tumour cells did not express PD-L1. In another case of the same tumour type [96], a combination of ipilimumab and novolumab was used at the second relapse (after radical surgery and radiation), which contributed to a complete and sustained response.

Another study [97] showed a durable response to a combination of ipilimumab and nivolumab in recurrent cervical NEC and no recurrence of the disease.

For small-cell ovarian cancer [98], tirelizumab was used as a combination of etoposide and cisplatin, which inhibited disease progression and stabilised the disease.

Based on the above data, it can be concluded that immunotherapy appears to be a promising form of treatment; however, this requires further research.

## 7. Chemotherapy

Various types of chemotherapeutics are used in the treatment of GEP NENs [99]. The most commonly used are combinations of cisplatin and etoposide. According to ESMO guidelines [24], the use of everolimus is recommended for progressive disease. Sunitinib is recommended in overtreated Pan-NETs. Cisplatin or carboplatin in combination with etoposide is recommended for NEC G3, but cisplatin or carboplatin should not be used in NET G3 (poor response to treatment). For the purpose of this article, the focus is on chemotherapeutics also used in NENs of the reproductive tract.

In the large BETTER clinical trial [100], bevacizumab, fluorouracil, and streptozocin were used as a treatment in the well-differentiated NETs of the pancreas at the metastatic stage. The mean progression-free survival was 23.7 months. A total of 56% of patients had a partial response to the treatment, while 44% demonstrated stabilisation of the disease.

In the RADIANT-4 clinical trial [101], everolimus was used in the treatment of lung or gastrointestinal NENs. The mean progression-free survival was 11 months in the everolimus group, while in the placebo group, it was 3.9. Additionally, the RADIANT-3 trial [102] investigated the efficacy of everolimus in advanced-stage pancreatic NETs. The mean survival time in the everolimus group was 44 months, while in the placebo group, it was 37.7 months. The mean progression-free survival was 11 months in the everolimus group and 4.6 months in the placebo group [103].

A recently published study [104] investigated the effect of biomarkers on the response to the treatment with etoposide and platinum in G3 NENs. A total of 56% of patients had gastrointestinal NENs. The response to the treatment was higher in NENs with Rb inap (63% vs. 42%), with similar results for NENs with p16 high (66% vs. 35%). Interestingly, in another study [105], Rb status had no effect on the treatment with platinum and etoposide. The TOPIC-NEC study [106] compared the efficacy between etoposide and platinum and between irinotecan and platinum. The mean overall survival in the etoposide group was 12.5 months, and 10.9 months in the irinotecan group. The mean progression-free survival was 5.6 months and 5.1 months, respectively.

The efficacy of the different chemotherapy mixtures (5-fluorouracil, irinotecan, and oxaliplatin) with bevacizumab as a fixed component was investigated [107].

For the whole group of patients, the response to the treatment reached 63.6%, and the mean progression-free survival and overall survival were 14 months and 15.3 months, respectively. In another study [108], a combination of bevacizumab and capecitabine was used in the treatment of GEP NETs. The mean progression-free survival was 24.3 months, and the mean overall survival was not reached. A similar result was obtained in another study [109] investigating the same drug mixtures.

Another study [110] investigated the efficacy of bevacizumab in a combination with the mTOR inhibitor temsirolimus in pancreatic NETs. The response rate was 41%, the mean progression-free survival was 13.2 months, and the mean overall survival was 34 months. These results appear to be quite promising.

The above data are briefly summarised in Table 2.

According to the Society of Gynaecologic Oncology guidelines [16], adjuvant chemotherapy—etoposide and platinum—in at least 5 cycles is recommended for early-stage cervical NEN (<IB3). It is also possible to use platinum with paclitaxel. In the advanced stages (IB3 and beyond), etoposide with platinum is also used due to its efficacy and lower toxicity compared to other chemotherapeutics. For ovarian NEC, it is recommended to consider the administration of an mTOR inhibitor (everolimus). Bevacizumab, pegylated interferon alfa, and tyrosine kinase inhibitors may also be considered.

Ishikawa et al. [18] investigated the efficacy of different chemotherapeutics in cervical NEN. The treatment response presented as follows: 46.3% in etoposide-platinum, 33.3% in platinum monotherapy, 39.1% in irinotecan-platinum, 12.9% in taxane-platinum, 50% in other platinum mixtures, and 5.9% in platinum-free mixtures. The overall response rate was higher in a high-grade NEC compared to a low-grade NEC (36.9% vs. 8.7%, respectively).

For small cell carcinomas of the cervix [69], the combination of etoposide and platinum showed superior efficacy to other chemotherapeutics, with progression-free survival (PFS) at 5 years of 67.6% vs. 20.9%, respectively. For the entire study group, the mean PFS was 39 months. In a small study group of nine cases of small-cell cervical cancer [111], the efficacy of etoposide and platinum was compared to vincristine, adriamycin, and cyclophosphamide. Five patients had early-stage (IB) staging. A total of 4 patients with advanced disease who received the VAC mixture had a response rate of 75%. After 5 years, in the whole group, the overall survival reached 52%, and the progression-free survival at 56%. In patients with an early stage of the disease, the overall survival after 5 years was 80%; in the advanced stage, it remained at only 25%.

Another study [112] examined 17 cases of small-cell cervical cancer and the efficacy of platinum-based chemotherapy. The patients were given either cisplatin or carboplatin in a combination with etoposide. The mean progression-free survival was 9.1 months for the whole group, 31.2 months in patients at an early stage of disease (IB2 inclusive), and 6.2 months in the advanced group. The overall survival at 3 years in patients who received chemotherapy in their initial treatment was 83, while only 20 in those who did not receive chemotherapy initially. In giant cell carcinoma [113], it was shown that both platinum-based chemotherapy or a combination with etoposide in an early stage (IB) was associated with a better prognosis.

The efficacy of the treatment of ovarian NET has also been investigated [114]. It was shown that in both early stage (I-II) and advanced stage (III-IV), 5-year survival after radical surgery was higher after concurrent chemoradiation (65.5% and 72.9%, respectively) than chemotherapy alone (39.3% and 21.8%, respectively). In a case report of giant cell ovarian NEC [115], the treatment with chemotherapy included paclitaxel and carboplatin. Despite the treatment, the patient died after 4 months. In contrast, another patient with the same tumour [109] remained in remission after the treatment with etoposide and cisplatin. In the treatment of primary aggressive NET of the ovary [116], everolimus was successfully used.

## 8. Conclusions

Both gastrointestinal and reproductive tract neuroendocrine tumours are rare tumours with poor prognoses. In recent years, the incidence of GEP NENs has increased as a result of the development of diagnostic methods. Not all risk factors are known. Although resection is the basis of treatment, in combination with other treatment methods, they can result in permanent remission. Non-surgical treatments are still developing, with one of the most rapidly growing being immunotherapy, which may prove to be a highly effective treatment option.

## Figures and Tables

**Figure 1 ijms-24-06853-f001:**
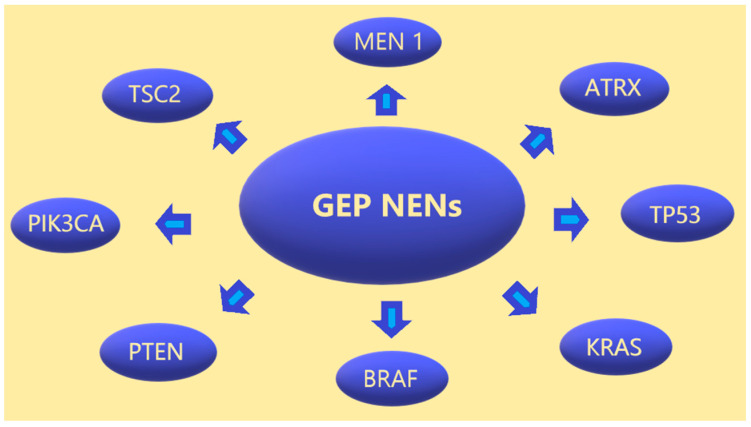
GEP NENs and their most common mutations.

**Figure 2 ijms-24-06853-f002:**
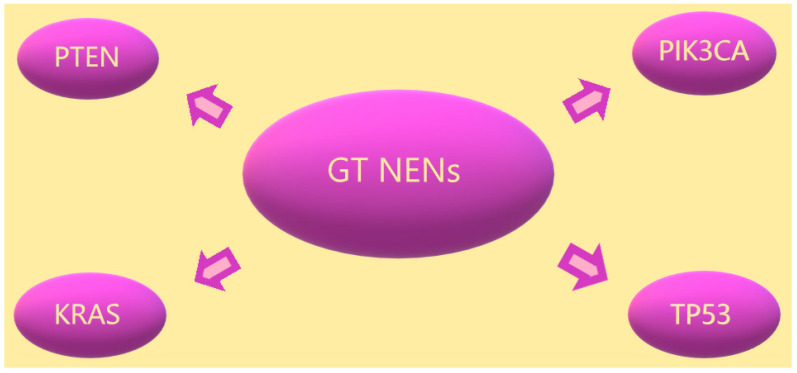
Gynaecological tract NENs and their most common mutations.

**Table 1 ijms-24-06853-t001:** Results of immunotherapy in GEP NENs.

Progression-Free Survival [Months]	Response Rate [%]	Overall Survival [Months]	Treatment	Study Group
nd	12 (PD-L1 positive) vs. 6.3(PD-L1 negative)	nd	pembrolizumab	KEYNOTE-028 [85]
4.1	3.7	24.2	pembrolizumab	KEYNOTE-158 [86]
1.9	19.1	nd	pembrolizumab	MacFarlane et al. [87]
4.5	24.1	5.1	pembrolizumab	Vijayvergia et al. [88]
2.1	20.8	10.8	sintilimab	Jia et al. [89]
31% (after 6 months)	25	11	Ipilimumab + nivolumab	Patel et al. [90]
4.8	24	14.8	Ipilimumab + nivolumab	Klein et al. [91]
2.5	20	7.8	toripalimab	Lu et al. [92]

nd: no data.

**Table 2 ijms-24-06853-t002:** Results of chemotherapy in GEP NENs.

Progression-Free Survival [Months]	Response Rate [%]	Overall Survival [Months]	Treatment	Study Group
23.7	56	21.1	bevacizumab	BETTER [100]
11	64	23.7	everolimus	RADIANT-4 [101]
nd	Nd	44	everolimus	RADIANT-3 [103]
5.9	60	12	Etoposide + platinum	Hadoux et al. [105]
14	63.6	15.3	bevacizumab	Collot et al. [107]
23.4	88	Not reached	Bevacizumab +capecitabine	BETTER [108]
13.2	41	34	Bevacizumab + temsirolimus	Hobday et al. [110]

nd: no data.

## Data Availability

Data sharing is not applicable.

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
