# Peer review of "Neuroendocrine Neoplasms of the Gastrointestinal Tract versus Neuroendocrine Neoplasms of the Gynaecological Tract—Comparison of the Risk Factors and Non-Surgical Treatment Efficacy"

_ijms, 2023, doi:10.3390/ijms24076853_

Round 1

Reviewer 1 Report

Neuroendocrine tumours of the gastrointestinal tract are rare. The incidence has increased in recent years due to improvements in diagnostic methods in detecting these lesions.

The situation is similar in even rarer neuroendocrine tumours of the reproductive tract, which are associated with an equally poor prognosis. 

Because of the heterogeneity of the neuroendocrine tumor and the endocrine function of the female reproductive system, the key to treatment is to differentiate neuroendocrine tumor from tumors of the female reproductive system, it is necessary for the author to further discuss the differential diagnosis of the two diseases and clarify their similarities and differences, so as to provide the basis for diagnosis and treatment.

Author Response

Dear Reviewer, 

in the attached document we send you our respond to your review of our article

Kind regards,

Anna Lorenz

Reviewer 2 Report

Interesting Title, extensive review of NEN RFs in general as well as genetics, but therapeutic review is not according to the actual management of NEN.

Author Response

Dear reviewer,

we send you our respond to your review of our article.

Kind regards,

Anna Lorenz

Round 2

Reviewer 2 Report

You have to describe therapeutics according to ENETS/ESMO/NANETS Guidelines.

New ENETS Guidance Papers are going to be published soon.

Author Response

Dear Reviewer,

I have attached our answer to your review.

Kind regards,

Anna Lorenz
